# Correlates of SARS-CoV-2 Variants on Deaths, Case Incidence and Case Fatality Ratio among the Continents for the Period of 1 December 2020 to 15 March 2021

**DOI:** 10.3390/genes12071061

**Published:** 2021-07-12

**Authors:** Wajdy J. Al-Awaida, Baker Jawabrah Al Hourani, Samer Swedan, Refat Nimer, Foad Alzoughool, Hamzeh J. Al-Ameer, Sara E. Al Tamam, Raghad Alashqar, Omar Al bawareed, Yulia Gushchina, Mohamed Samy Abousenna, Amani Marwan Ayyash, Ahmad Sharab, Sulaiman M. Alnaimat, Amer Imraish, Muhanad W. Akash, Najah R. Hadi

**Affiliations:** 1Department of Biology and Biotechnology, Faculty of Science, American University of Madaba, Madaba 11821, Jordan; h.alameer@aum.edu.jo (H.J.A.-A.); sarattt2001148@gmail.com (S.E.A.T.); 1910332@std.aum.edu.jo (R.A.); a.sharab@aum.edu.jo (A.S.); 2Department of Biology and Chemistry, Embry-Riddle Aeronautical University, Prescott, AZ 86301, USA; Jawabrab@erau.edu; 3Department of Medical Laboratory Sciences, Jordan University of Science and Technology, Irbid 22110, Jordan; sfswedan4@just.edu.jo (S.S.); rmnimer@just.edu.jo (R.N.); 4Department of Medical Laboratory Sciences, The Hashemite University, Zarqa 13133, Jordan; foad@hu.edu.jo; 5Department of Normal Physiology, Peoples’ Friendship University of Russia (RUDN University), 6 Miklukho-Maklaya Street, 117198 Moscow, Russia; alomar442@mail.ru; 6Department of General and Clinical Pharmacology, Peoples’ Friendship University of Russia (RUDN University), 6 Miklukho-Maklaya Street, 117198 Moscow, Russia; gushchina-yush@rudn.ru; 7Central Laboratory for Evaluation of Veterinary Biologics, Agricultural Research Center, Cairo 11517, Egypt; mohamedsamy2020@hotmail.com; 8Department of Pharmacy, Faculty of Heath Sciences, American University of Madaba, Madaba 11821, Jordan; a.ayyash@aum.edu.jo; 9Biological Department, Faculty of Science, Al Hussein Bin Talal University, Ma’an P.O. Box 20, Jordan; s_alnaimat@ahu.edu.jo; 10Department of Biology, Faculty of Science, The University of Jordan, Amman 11942, Jordan; a.imraish@ju.edu.jo; 11Department of Horticulture and Crop Science, Faculty of Agriculture, The University of Jordan, Amman 11942, Jordan; makash@ju.edu.jo; 12Department of Pharmacology and Therapeutics, Faculty of Medicine, University of Kufa, Najaf 540011, Iraq; drnajahhadi@yahoo.com

**Keywords:** SARS-CoV-2, COVID-19, GISAID, deaths per million, cases per million, Spearman’s correlation, genomic variations, case fatality ratio

## Abstract

The outbreak of coronavirus disease 2019 (COVID-19), by the severe acute respiratory syndrome coronavirus 2 (SARS-CoV-2), has quickly developed into a worldwide pandemic. Mutations in the SARS-CoV-2 genome may affect various aspects of the disease including fatality ratio. In this study, 553,518 SARS-CoV-2 genome sequences isolated from patients from continents for the period 1 December 2020 to 15 March 2021 were comprehensively analyzed and a total of 82 mutations were identified concerning the reference sequence. In addition, associations between the mutations and the case fatality ratio (CFR), cases per million and deaths per million, were examined. The mutations having the highest frequencies among different continents were Spike_D614G and NSP12_P323L. Among the identified mutations, NSP2_T153M, NSP14_I42V and Spike_L18F mutations showed a positive correlation to CFR. While the NSP13_Y541C, NSP3_T73I and NSP3_Q180H mutations demonstrated a negative correlation to CFR. The Spike_D614G and NSP12_P323L mutations showed a positive correlation to deaths per million. The NSP3_T1198K, NS8_L84S and NSP12_A97V mutations showed a significant negative correlation to deaths per million. The NSP12_P323L and Spike_D614G mutations showed a positive correlation to the number of cases per million. In contrast, NS8_L84S and NSP12_A97V mutations showed a negative correlation to the number of cases per million. In addition, among the identified clades, none showed a significant correlation to CFR. The G, GR, GV, S clades showed a significant positive correlation to deaths per million. The GR and S clades showed a positive correlation to number of cases per million. The clades having the highest frequencies among continents were G, followed by GH and GR. These findings should be taken into consideration during epidemiological surveys of the virus and vaccine development.

## 1. Introduction

The novel betacoronavirus, severe acute respiratory syndrome coronavirus 2 (SARS-CoV-2), responsible for coronavirus disease 2019 (COVID-19), was first described in Wuhan, China in December 2019 [1,2,3]. Subsequently, COVID-19 became a worldwide pandemic. According to the World Health Organization (WHO), for the period of 1 December 2020 to 15 March 2021, the worldwide outbreak of COVID-19 has approximately 64,668,248 cases and more than 1,339,685 deaths were reported. SARS-CoV-2 is an enveloped single-stranded positive RNA virus [4]. The SARS-CoV-2 complete genome consists of about 30,000 nucleotides that translate structural and non-structural proteins (Nsps) [1,2]. Coronaviruses share a roughly spherical or moderately pleomorphic shape with distinct spike (S) protein envelope projections [5]. The four SARS-CoV-2 structural proteins are the spike (S), the envelope (E), the membrane (M) and the nucleocapsid (N) proteins, while nonstructural proteins are 3-chymotrypsin-like protease (nsp3 and nsp5), papain-like protease and RNA-dependent RNA polymerase (nsp12) [6,7]. Surface S glycoprotein attaches to the host’s angiotensin-converting enzyme 2 (ACE2) receptor and plays an important role in viral adsorption and human transmission [8]. Nsps, created as cleavage products from open reading frame 1ab (ORF1ab), are assembled to facilitate transcription and viral replication. Viral RNA is synthesized and controlled by the RNA-dependent polymerase with assistance from Nsp7 and Nsp88 [9].

SARS-CoV-2 genetic mutations may allow the virus to adapt to a new host cell or alter virus pathogenesis [10]. Therefore, detecting and analyzing virus mutations in different countries could enhance our understanding of the contribution of such mutations to viral pathogenesis and spread. In contrast to SARS-CoV in 2002 and Middle East Coronavirus Respiratory Syndrome (MERS-CoV) in 2012, SARS-CoV-2 spread worldwide quickly. To date, the greatest assessment of the fatality ratio of SARS-CoV-2 has come from data recorded in China, France and the Diamond Princess cruise ship [11]. Effective vaccines and treatments are urgently needed to decrease the mortality and morbidity ratio due to COVID-19. Variations have been observed in mortality ratios from country to country [3]. However, factors explaining these variations have not been elucidated. Furthermore, the contribution of SARS-CoV-2 genetic variations and host genetic factors to mortality ratio remains unknown.

In this study, we evaluated 553,518 SARS-CoV-2 genome sequences isolated from different continents and explored their associations with CFR, cases per million and deaths per million.

## 2. Materials and Methods

### 2.1. Data Availability

Full-length viral nucleotide sequence of the reference SARS-CoV-2 Wuhan-Hu-1 strain (Accession NC_045512) [1] was obtained from the global initiative on sharing all influenza data (GISAID). We used a total of 553,518 SARS-CoV-2 genome sequences isolated from patients/individuals in different continents, including 157,691 from North America, 6702 from South America, 346,951 from Europe, 1670 from Australia, 34,736 from Asia and 5768 from Africa. SARS-CoV-2 genome sequences were collected from a publicly open database, the GISAID (https://www.gisaid.org/ (accessed on 1 December 2020)).

### 2.2. Mutation Analysis

Only high-coverage SARS-CoV-2 sequences were downloaded from GISAID EpiCoVTM database [12]. Each of the SARS-CoV-2 sequences was compared with the reference sequence SARS-CoV-2 Wuhan-Hu-1 (accession number MN908947.3) by using CoVsurver enabled by GISAID (https://www.gisaid.org/epiflu-applications/covsurvermutations-app/ (accessed on 1 December 2020)) [12,13]. After the alignment of the full-length viral nucleotide sequence, we identified SARS-CoV-2 mutations in each of the isolated virus sequences. Single mutations based on a population average were used for the estimation of the CFR.

Genomic clades were inferred by GISAID database and defined according to its nomenclature system at the time of data collection outlined in (https://www.gisaid.org/references/statements-clarifications/clade-and-lineage-nomenclature-aids-in-genomic-epidemiology-of-active-hcov-19-viruses/ (accessed on 1 December 2020)).

### 2.3. Data Acquisition

The number of deaths and confirmed cases related to COVID-19 was collected from the Worldometer (https://www.wfiveorldometers.info/coronavirus/ (accessed on 1 December 2020)). The CFR was calculated by dividing total deaths by total confirmed cases in each country/region.

### 2.4. Phylogenetic Analysis

In comparison with the reference sequence of SARS-CoV-2 Wuhan-Hu-1 strain (Accession NC_045512.2), phylogenetic analysis was performed for the complete coding region of all isolates using Nextstrain software [14].

### 2.5. Ethical Statement

The study was carried out following the Helsinki Declaration. Ethical approval was obtained from the Institutional Review Board at the American University of Madaba (Ref. H20010).

### 2.6. Statistical Analyses

The data were analyzed using the Social Sciences Statistical Package version 23 (IBM Inc., Armonk, New York, NY, USA). Hierarchical cluster analysis was used to identify mutant clusters among the countries/regions. Spearman’s correlation (r) values and two-tailed significance (*p*) were used to identify the correlation between mutation frequencies with CFR, cases per million and deaths per million. A *p*-value of less than 0.05 was considered to be statistically significant.

## 3. Results

COVID-19 related information for different continents is shown in Table 1. Africa had the highest CFR of 2.56%; Australia had the lowest CFR of 0.79% (Table 1). The highest recorded number of cases per million in Europe (58,437), while the lowest was in Africa (6295). The highest recorded number of deaths per million was in Europe (1117), while the lowest was in Australia (45).

### 3.1. Clades Frequencies of SARS-CoV-2 Genome

The clades for 553,518 SARS-CoV-2 genome sequences from different world continents were identified. The clades frequencies are shown in Figure 1A. The clades having the highest frequencies among continents were G, GH and GR. The highest frequencies for G clade were in South America and the predominance of clade GH in North America. The clades GR predominated in Asia (Appendix A).

The cluster analysis of the mutations indicates similarity among SARS-CoV-2 isolates from South America and Asia; isolates from Australia have a similarity to those in America and Asia and a lesser degree similarity among SARS-CoV-2 isolates from Europe and Africa. On the other hand, isolates from North America were least similar to those from other regions (Figure 1B).

### 3.2. Correlation between SARS-CoV-2 Genetic Clades with Case Fatality Ratio (CFR), Cases per Million and Deaths per Million

Table 2 demonstrates Spearman’s correlation (r) values and two-tailed significance (*p*) values between the SARS-CoV-2 clades and each of the CFR, cases per million and deaths per million. Among the identified clades, none showed a significant correlation to CFR. The G, GR, GV, S clades showed a significant positive correlation to deaths per million. The GR and S clades showed a significant positive correlation to number of cases per million.

### 3.3. Correlation between SARS-CoV-2 Genetic Variants with Case Fatality Ratio (CFR), Cases per Million and Deaths per Million

Appendix A demonstrates Spearman’s correlation (r) values and two-tailed significance (*p*) values between the SARS-CoV-2 mutations and each of the CFR, cases per million and deaths per million. Among the identified mutations, NSP2_T153M, NSP14_I42V and Spike_L18F mutations showed a positive correlation to CFR (Figure 2A–C). However, NSP3_T73I and NSP3_Q180H, NSP13_Y541C showed a negative correlation to CFR (Figure 3A–C). Spike_D614G and NSP12_P323L mutations displayed a positive correlation to deaths per million (Figure 4A,B). The NSP3_T1198K, NS8_L84S and NSP12_A97V mutations showed a significant negative correlation to deaths per million (Figure 4C–E). NSP12_P323L and Spike_D614G mutations showed a positive correlation to number of cases per million (Figure 5A,B). However, NS8_L84S and NSP12_A97V mutations showed a negative correlation to the number of Cases per million (Figure 5C,D).

### 3.4. Global Mapping of Frequencies of the Mutations Correlating with CFR, Cases per Million and Deaths per Million among World Continents

Appendix A demonstrates the clustering of SARS-CoV-2 isolates and heatmap of the frequency of mutations of the SARS-CoV-2 genome by continent. The NSP2_T153M mutation was reported at 1.18%, 0.02%, 0.04%, 0.07% and 0.07% in SARS-CoV-2 genome sequences isolated from Africa, Asia, Europe, North America and South America, respectively (Figure 2D). The NSP14_I42V mutation was reported at 0.017 and 0.003 in SARS-CoV-2 genome sequences isolated from Africa and Europe, respectively (Figure 2E). The Spike_L18F mutation was reported at 16.02%, 0.72%, 1.86%, 10.90%, 1.25% and 5.94% in SARS-CoV-2 genome sequences isolated from Africa, Asia, Australia, Europe, North America and South America, respectively (Figure 2F).

The NSP3_T73I mutation was reported at 0.02%, 0.05%, 0.18%, 0.05%, 0.10% and 0.03% in SARS-CoV-2 genome sequences isolated from Africa, Asia, Australia, Europe, North America and South America, respectively (Figure 3D). The NSP3_Q180H mutation was reported at 0.05%, 0.15%, 0.18%, 0.08%, 1.18% and 0.10% in SARS-CoV-2 genome sequences isolated from Africa, Asia, Australia, Europe, North America and South America, respectively (Figure 3E). The NSP13_Y541C mutation was reported at 0.01%, 0.84%, 0.01%, 0.15% and 0.04% in SARS-CoV-2 genome sequences isolated from Asia, Australia, Europe, North America and South America, respectively (Figure 3F).

The Spike_D614G mutation was reported at 91.97%, 96.46%, 91.02%, 98.99%, 97.91% and 97.02% in SARS-CoV-2 genome sequences isolated from Africa, Asia, Australia, Europe, North America and South America, respectively (Figure 4F and Figure 5F). The NSP12_P323L mutation was reported at 88.66%, 95.52%, 91.08%, 98.35%, 94.79% and 96.43% in the SARS-CoV-2 genome sequences isolated from Africa, Asia, Australia, Europe, North America and South America, respectively (Figure 4G and Figure 5E). The NSP3_T1198K mutation was reported at 0.21%, 1.56%, 0.18%, 0.01%, 0.02% and 0.07% in SARS-CoV-2 genome sequences isolated from Africa, Asia, Australia, Europe, North America and South America, respectively (Figure 4H). The NS8_L84S mutation was reported at 4.09%, 1.39%, 3.29%, 0.25%, 0.51% and 0.25% in SARS-CoV-2 genome sequences isolated from Africa, Asia, Australia, Europe, North America and South America, respectively (Figure 4I and Figure 5G). The NSP12_A97V mutation was reported at 0.28%, 0.52%, 1.98%, 0.05%, 0.23% and 0.09% % in SARS-CoV-2 genome sequences isolated from Africa, Asia, Australia, Europe, North America and South America, respectively (Figure 4J and Figure 5H).

Other mutations have a frequency higher than 20% without having a significant correlation with CFR, cases per million and deaths per million among world continents. The N_G204R mutation was reported at 22.31%, 75.69%, 46.59%, 52.36% and 76.01% in SARS-CoV-2 genome sequences isolated from Africa, Asia, Australia, Europe and South America, respectively. The N_R203K mutation was reported at 24.24%, 75.88%, 47.90%, 53.51%, 15.08% and 76.41% in SARS-CoV-2 genome sequences isolated from Africa, Asia, Australia, Europe, North America and South America, respectively. The NSP2_T85I mutation was reported at 40.20% and 59.59% in SARS-CoV-2 genome sequences isolated from Africa and South America, respectively. The NS3_Q57H mutation was reported at 46.10%, 32.99% and 63.30% in SARS-CoV-2 genome sequences isolated from Africa, Australia and North America, respectively. The Spike_E484K mutation was reported at 35.63% in SARS-CoV-2 genome sequences isolated from Africa. The Spike_N501Y mutation was reported at 36.55% and 47.23% in SARS-CoV-2 genome sequences isolated from Africa and Europe, respectively. The Spike_H69del mutation was reported at 48.93% in SARS-CoV-2 genome sequences isolated from Europe. The Spike_K417N mutation was reported at 34.36% in SARS-CoV-2 genome sequences isolated from Africa. The NS8_S24L mutation was reported at 33.88% in SARS-CoV-2 genome sequences isolated from North America. The NS8_Q27stop mutation was reported at 46.36% in SARS-CoV-2 genome sequences isolated from Europe. The Spike_A222V mutation was reported at 26.59% in SARS-CoV-2 genome sequences isolated from Europe. The Spike_V1176F mutation was reported at 29.30% in SARS-CoV-2 genome sequences isolated from South America. The N_I292T mutation was reported at 20.50% in SARS-CoV-2 genome sequences isolated from South America. The NS3_S171L mutation was reported at 30.37% in SARS-CoV-2 genome sequences isolated from Africa.

### 3.5. Phylogenetic Clades Tree of SARS-CoV-2 in Different World Continents

The phylogenetic analysis indicated that several clades were present on different continents. The analysis also revealed that clades 20 became dominant. Clades 20 (including 20A, 20B, 20C, 20E (EU1), 20F, 20G, 20H/501Y.V2, 20I/501Y.V1 and 20J/501Y.V2) by Nextstrain nomenclature, are also known as clades G, GR and GH by GISAID nomenclature (Figure 6).

## 4. Discussion

The current COVID-19 outbreak has spread rapidly worldwide. Most COVID-19 infected patients suffering from COVID-19 are either asymptomatic or experience mild symptoms. Around 15% of all COVID-19 patients progress to severe pneumonia and approximately 5% of COVID-19 patients have acute respiratory disorder syndrome, septic shock and multiple organ failure [15,16]. The CFR associated with COVID-19 differs from continent to continent and is higher in Africa, Europe, South America and North America countries compared with those in Asia and Australia. While several hypotheses have been put forward to explain the variations in CFR, including age distribution differences, blood groups, gender, virus genomic types and ethnic backgrounds, many of these hypotheses still require confirmation.

This study explored and compared mutation profiles of SARS-CoV-2 isolates from different continents. We identified differences among continents relating to COVID-19 CFR, cases per million and deaths per million. The differences may be attributed to differences in age distributions [17,18], blood groups [19], gender [20,21], virus genomic types [3], strict lockdown strategies [22] social distancing [23] and genetic backgrounds [24,25]. Hence, this study also investigated the contribution of genetic variations in the SARS-CoV-2 genome to the COVID-19 CFR, the number of cases per million and the number of deaths per million.

Like other RNA viruses, genetic variation in SARS-CoV-2 is important for fitness, survival and pathogenesis. Spontaneous mutations and recombination are two main sources of genetic variation for SARS-CoV-2 [26].

Based on data obtained from a public database GISAID for the period 1 December 2020 to 15 March 2021, the mutation rate of the SARS-CoV-2 virus is about 8 × 10^−4^ nucleotides/genome per year [27], which is considered a high rate for an RNA virus [27,28,29]. Analysis of 553,518 genomic sequences from the GISAID databases showed that the number and the frequency of mutations in Africa and Europe were considerably higher than in Asia [30]. The identification of SARS-CoV-2 mutation patterns indicated differences relating to geography, time of isolation and subject age, but not relating to gender [31].

The mutation is one of the most effective evolutionary mechanisms in RNA viruses [32]. SARS-CoV-2 genomic studies described mutations in several genes, including ORF1ab, ORF3a N, S, M, E, ORF8, ORF7, ORF6 and ORF10 [33,34,35]. The S, ORF1ab, NSP15, NSP12, NSP3, NSP2, NSP1 and ORF8 genes had significantly higher mutation frequencies than other genes [33,36,37]. Several studies have been carried out to identify genomic variants of SARS-CoV-2, including synonymous, nonsynonymous, deletion, addition and non-coding mutations [31,33,38]. The most common mutations among the SARS-CoV-2 genome were nonsynonymous and synonymous mutations [31,36]. In the current study, the mutations having the highest frequencies among continents were missense, including Spike_D614G and NSP12_P323L. Moreover, the highest frequencies for Spike_D614G and NSP12_P323L mutations were among isolates from Europe with 98.99% and 98.35%, respectively.

Our study indicated a significant correlation between the number of both the number of cases per million and the number of deaths per million, with the Spike_D614G mutation, which is a mutation outside the receptor-binding domain (RBD). In line with previous studies, this study revealed that the Spike_D614G mutation showed a positive correlation with viral infectivity and enhanced transmissibility [39,40,41].

The genome of SARS-CoV-2 is about 30 kb and codes for 14 open frames (ORFs). 5′-ORF-1a/1b forms 67% of the genome and encodes polyproteins 1a (pp1a) and polyproteins 1 ab (pp1ab), which are then processed into sixteen non-structural proteins (nsp1-16) [42]. Other nsp proteins are believed to be associated with viral replication or host immune response modulation [43]. In our study, many genetic variants in non-structural proteins, including NSP3_T73I, NSP3_Q180H and NSP13_Y541C mutations showed a significant negative correlation to CFR. In addition, NSP3_T1198K, NS8_L84S and NSP12_A97V mutations showed a significant negative correlation to the deaths per million. NSP12_P323L mutation showed a positive correlation to deaths per million. The NS8_L84S and NSP12_A97V mutations showed a negative correlation to number of cases per million. In addition, the NSP1_M85I, NSP3_E475D showed a significant positive correlation to the number of cases per million. These mutations may affect viral replication or host immune response modulation, leading to change in viral pathogenicity [44].

The trimetric viral spike (S) glycoprotein is the main determinant of coronaviruses’ host specificity and pathogenesis. The S2 domain activates the viral and host cell membranes’ fusion to encourage viral penetration and uncoating [45,46]. In our study, The Spike_L18F mutation showed a significant positive correlation to CFR. On the other hand, The Spike_D614G mutation showed a significant positive correlation to the number of cases per million and the number of deaths per million. According to our findings, The Spike_D614G mutation in the S2 domain has a positive correlation between viral infectivity and transmissibility.

A recent report have shown that the crystal structure of the SARS–CoV-2 nucleocapsid (N) protein is identical to that of the known coronavirus N proteins, but their surface electrostatic potential properties were distinct [47]. Sheik et al. examined the variables impacting the variation of the N gene among 13 coronaviruses, as well as how they affect the virus-host relationship, reporting high AT% and low GC% in the coronavirus SARS N gene [48]. The N protein of coronaviruses plays a major role in the packaging of the viral genome and virus assembly [49]. In our study, NS8_L84S mutation showed a significant negative correlation to deaths per. These mutations may have a critical role in multiple steps of the viral life cycle.

The phylogenetic analysis indicated that several virus clades were deployed in different continents and it was observed that clades 20 became dominant. Clades 20 included 20A, 20B, 20C, 20E(EU1), 20F, 20G, 20H/501Y.V2, 20I/501Y.V1 and 20J/501Y.V2 by Nextstrain nomenclature, which is also known as clades G, GR and GH by GISAID nomenclature [50]. The majority of genomes were classified into one of seven major clades: L, S, V, G, GH, GR, or GV [51,52,53]. The clades G, GH, GR and GV, which account for 98% of the genomes, were found to have the D614G mutation. Clade G was identified the most frequently, followed by GR and GH. The mutation was discovered in a glycosylated residue of the viral spike, which is conserved throughout this species [54]. Mutations in this area could be associated with alterations in host cell membrane fusion capacity [53], an effect that should also result in increased transmission and pathogenicity between individuals. Korber and colleagues subsequently provided experimental evidence that associate this mutation in COVID-19 patients with increased infectivity and greater viral loads [39]. Sub-clusters of clade G began to evolve into clades GH, GR and, more recently, clade GV. The current study’s analysis of the chronological distribution of SARS-COV-2 clades revealed a significant increase in the number of sequenced genomes clustered into the GR clade relative to clade G. Additionally, a change in the number of genomes clustered into clade GH was observed. Finally, on the basis of the details presented, the adaptation-driven genetic evolution hypothesis appears to be more viable. However, experimental evidence designed to allow for comparison of clades needs to be established.

## 5. Conclusions

The mutations having the highest frequencies among continents were Spike_D614G and NSP12_P323L. Among the identified mutations, NSP2_T153M, NSP14_I42V and Spike_L18F mutations showed positive correlation to CFR, while NSP13_Y541C, NSP3_T73I and NSP3_Q180H showed a negative correlation. The Spike_D614G and NSP12_P323L mutations showed positive correlation to deaths per million, while NSP3_T1198K, NS8_L84S and NSP12_A97V mutations showed negative correlation. NSP12_P323L and Spike_D614G mutations showed positive correlation to number of cases per million, while NS8_L84S and NSP12_A97V mutations showed negative correlation. In addition, among the identified clades, none showed a significant correlation to CFR. The G, GR, GV, S clades showed a significant positive correlation to deaths per million. The GR and S clades showed a positive correlation to number of cases per million. The clades having the highest frequencies among continents were G, followed by GH and GR. Further confirmation and biological significance studies are needed to verify association of identified mutations and clades with a CFR, number of deaths per million and number of cases per million

## 6. Limitations

Case fatality ratios, number of deaths per million and number of cases per million may be affected by several factors including age distribution differences, blood groups, gender, virus genomic types and ethnic backgrounds; many of these factors still require confirmation. 

## Figures and Tables

**Figure 1 genes-12-01061-f001:**
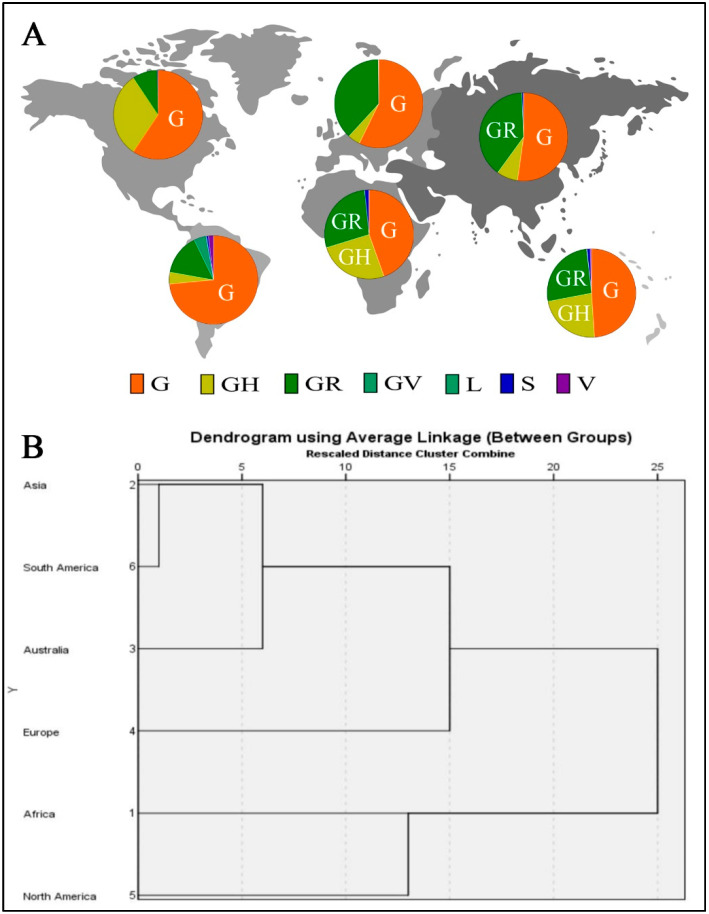
(**A**) Continent distribution of various SARS-CoV-2 clades. (**B**) Dendrogram showing clustering of SARS-CoV-2 isolates from different continents.

**Figure 2 genes-12-01061-f002:**
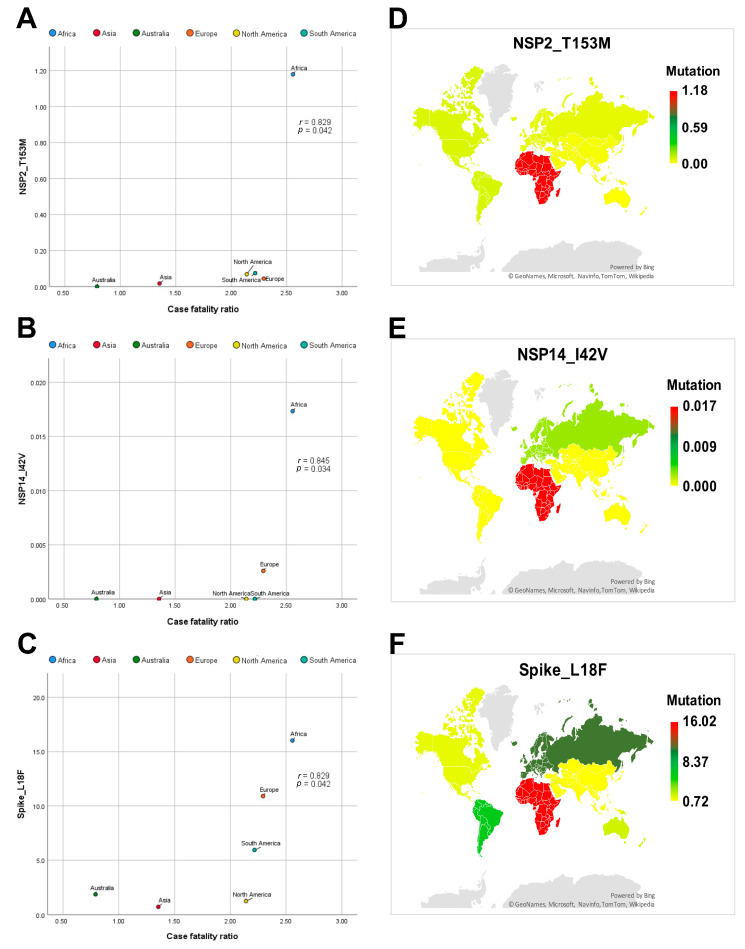
Correlation analysis and global mapping of mutations having a positive correlation with CFR among different world continents. (**A**) Spearman’s correlation between NSP2_T153M mutation and CFR among different world continents, (**B**) Spearman’s correlation between NSP14_I42V mutation and CFR among different world continents, (**C**) Spearman’s correlation between Spike_L18F mutation and CFR among different world continents, (**D**) Global mapping of NSP2_T153M mutation having a positive correlation with CFR among different world continents. (**E**) Global mapping of NSP14_I42V mutation having a positive correlation with CFR among different world continents. (**F**) Global mapping of Spike_L18F mutation having a positive correlation with CFR among different world continents r: Spearman’s correlation coefficient. *p*: *p*-value.

**Figure 3 genes-12-01061-f003:**
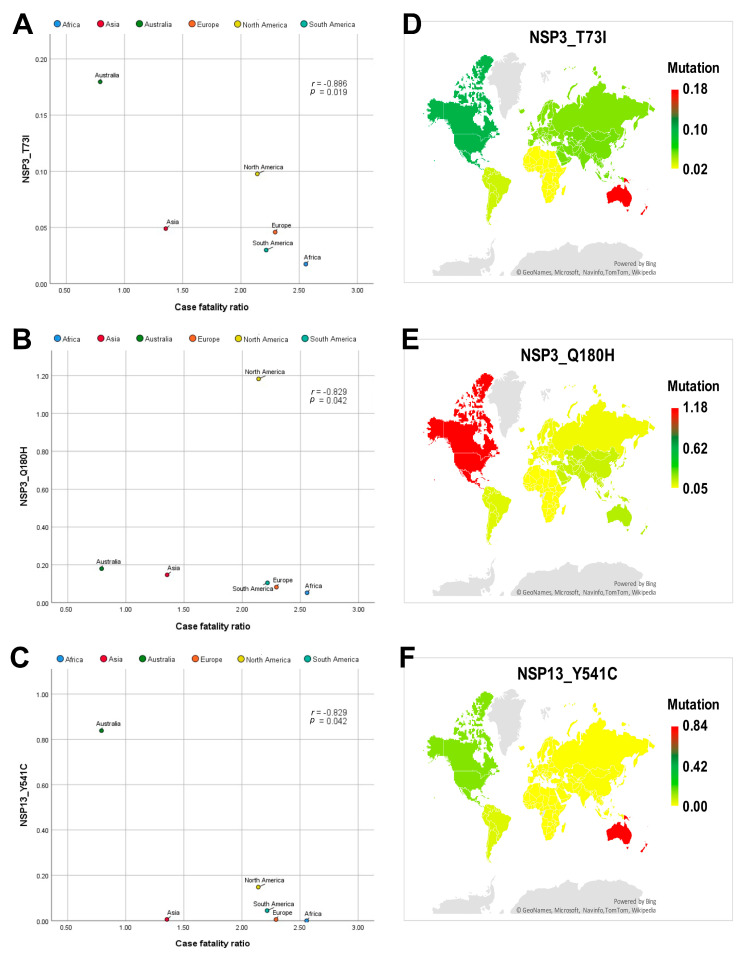
Correlation analysis and global mapping of mutations having a negative correlation with CFR among different world continents. (**A**) Spearman’s correlation between NSP3_T73I mutation and CFR among different world continents, (**B**) Spearman’s correlation between NSP3_Q180H mutation and CFR among different world continents, (**C**) Spearman’s correlation between NSP13_Y541C mutation and CFR among different world continents, (**D**) Global mapping of NSP3_T73I mutation having a negative correlation with CFR among different world continents. (**E**) Global mapping of NSP3_Q180H mutation having a negative correlation with CFR among different world continents. (**F**) Global mapping of NSP13_Y541C mutation having a negative correlation with CFR among different world continents r: Spearman’s correlation coefficient. *p*: *p*-value.

**Figure 4 genes-12-01061-f004:**
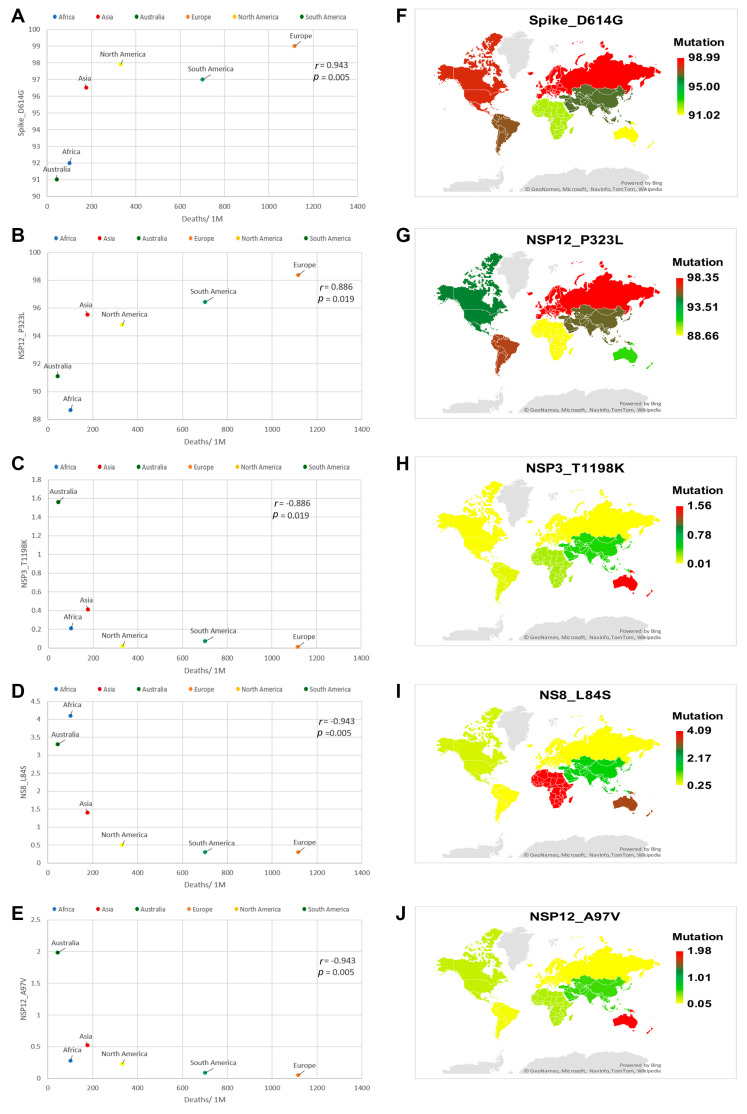
Correlation analysis and global mapping of mutations having a positive correlation with deaths per million among different world continents. (**A**) Spearman’s correlation between Spike_D614G mutation and deaths per million, (**B**) Spearman’s correlation between NSP12_P323L mutation and deaths per million, (**C**) Spearman’s correlation between NSP3_T1198K mutation and deaths per million. (**D**) Spearman’s correlation between NS8_L84S mutation and deaths per million. (**E**) Global mapping of NSP12_A97V mutation having a positive correlation with deaths per million. (**F**) Global mapping of Spike_D614G mutation having a positive correlation with deaths per million. (**G**) Global mapping of NSP12_P323L mutation having a positive correlation with deaths per million. (**H**) Global mapping of NSP3_T1198K mutation having a negative correlation with deaths per million. (**I**) Global mapping of NS8_L84S mutation having a negative correlation with deaths per million. (**J**) Global mapping of NSP12_A97V mutation having a negative correlation with deaths per million r: Spearman’s correlation coefficient. *p*: *p*-value.

**Figure 5 genes-12-01061-f005:**
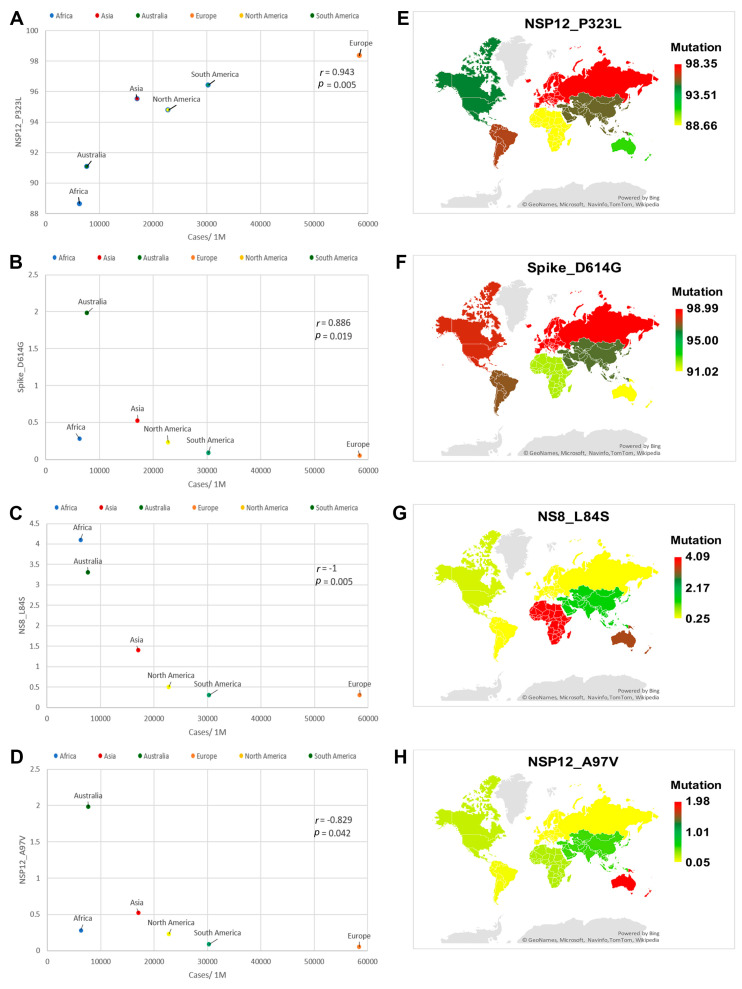
Correlation analysis and global mapping of mutations with cases per million among different world continents. (**A**) Spearman’s correlation between NSP12_P323L mutation and cases per million, (**B**) Spearman’s correlation between Spike_D614G mutation and cases per million (**C**) Spearman’s correlation between NS8_L84S mutation and cases per million. (**D**) Spearman’s correlation between NSP12_A97V mutation and cases per million. (**E**) Global mapping of NSP12_P323L mutation having a positive correlation with cases per million. (**F**) Global mapping of Spike_D614G mutation having a positive correlation with cases per million. (**G**) Global mapping of NS8_L84S mutation having a negative correlation with cases per million. (**H**) Global mapping of NSP12_A97V mutation having a negative correlation with cases per million. r: Spearman’s correlation coefficient. *p*: *p*-value.

**Figure 6 genes-12-01061-f006:**
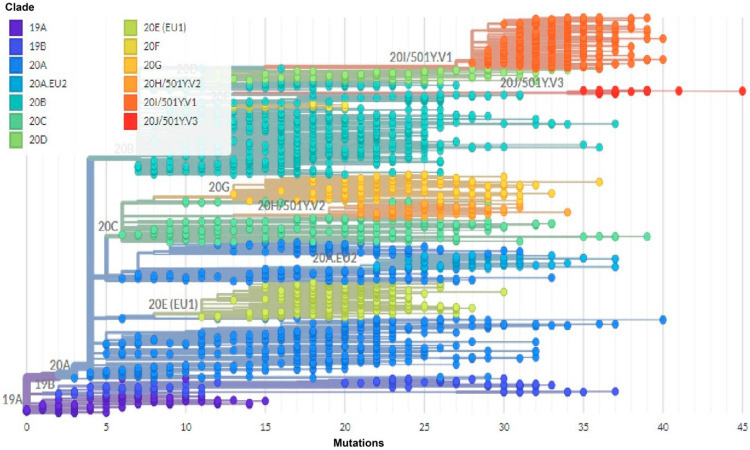
Phylogenetic analysis of SARS-CoV-2 isolates from different continents. SARS-CoV-2 clades were present in different continents.

**Table 1 genes-12-01061-t001:** COVID-19 statistics in the different world continents.

Region	Population	Total Cases	Cases/1M	Total Deaths	Deaths/1M	Case Fatality Ratio	Virus Isolates
Africa	1,363,047,821	3,067,758	6295	78,435	101	2.56	5768
Asia	4,624,453,127	12,546,291	17,067	169,912	177	1.35	34,736
Australia	42,643,478	24,777	7705	196	45	0.79	1670
Europe	747,950,109	23,239,185	58,437	532,996	1117	2.29	346,951
North America	592,362,766	17,757,444	22,816	380,088	333	2.14	157,691
South America	433,264,918	8,032,793	30,293	178,058	701	2.22	6702

**Table 2 genes-12-01061-t002:** Correlation analysis between SARS-CoV-2 clades and Case fatality rates, cases per million and deaths per million among the identified clades in all world continents.

Clade	Case Fatality Ratio		Deaths/1M		Cases/1M	
	r	*p*	r	*p*	r	*p*
Clade G	0.257	0.623	0.943	0.005 **	1	-
Clade GH	0.086	0.872	0.6	0.2	0.771	0.072
Clade GR	0.371	0.468	0.829	0.042 *	0.943	0.005 **
Clade GV	0.371	0.468	0.829	0.042 *	0.657	0.156
Clade GL	0.086	0.872	0.771	0.072	0.6	0.208
Clade S	0.371	0.468	0.829	0.042 *	0.943	0.005 **
Clade V	0.086	0.872	0.771	0.072	0.6	0.208

* Correlation is significant at the 0.05 level (2-tailed). ** Correlation is significant at the 0.01 level (2-tailed). r: Spearman’s correlation. *p*: *p*-value.

## Data Availability

Data is contained within the article or Appendix A.

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
