# Peer review of "Correlates of SARS-CoV-2 Variants on Deaths, Case Incidence and Case Fatality Ratio among the Continents for the Period of 1 December 2020 to 15 March 2021"

_genes, 2021, doi:10.3390/genes12071061_

Round 1

Reviewer 1 Report

Title of the manuscript: Correlates of SARS-CoV-2 variants on deaths, case incidence, and case fatality ratio among the continents for the period of December 1, 2020 to March 15, 2021

Manuscript ID: genes-1244710

Evaluation Summary: Al-Awaida WJ et al., has assessed the impact of mutations in SARS-CoV-2 genome with respect to fatality and also analyzed association of these mutations with case fatality ratio (CFR), cases per million and deaths per million by correlation. Comprehensive analysis by Al-Awaida WJ et al., identified 82 mutations with reference to wildtype genome (Wuhan strain) of SARS-CoV-2 while, mutations NSP2_T153M, NSP14_I42V, and Spike_L18F mutations showed a positive correlation to CFR and spike_D614G, NSP12_P323L mutations showed a positive correlation to deaths per million. Overall, authors have tried to provide an evidence on an important issue in the field, where SARS-CoV-2 variants/VOC’s have raised an alarming concern in various continents. 

Recommendations/Comments to authors:

Authors have extensively analyzed a huge number of viral genomes and correlated mutations with CFR and deaths per million. 

Major:

  1. Figure1 A & B: I recommend authors to make heatmap for the frequencies of SARS-CoV-2 mutants, classify them into clusters based on the mutational signature by a hierarchical clustering rather than just making stacking bar graphs. It is hard to understand that many colors.

  1. Though authors have extensively analyzed data by correlating single mutational to global CFR and deaths per million for figures 3, 4 and 5: Instead, it would be more informative and interesting to see correlations based on clusters or clades (like in figure6) obtained from hierarchical clustering which makes the entire manuscript simple and will be easy for the readers to understand.

  1. As D614G spike mutation in virus caused higher transmissibility and this mutation is detected in Europe in the early phase and has widely spread around the globe, especially to European and North American countries, it is obvious that it will be positively correlated with number of cases per million and deaths per million.

  1. I recommend authors to move table 2 to supplementary data once the heatmap is made.

Author Response

Response to Reviewers

We would like to thank the reviewers for careful and thorough review of this manuscript and for the thoughtful comments and constructive suggestions, which help to improve the quality of this manuscript.

Please find attached a point-by-point response to reviewer’s concerns. We hope that you find our responses satisfactory and that the manuscript is now acceptable for publication.

Response to Reviewer 1 Comments:

  1. Figure1 A & B: I recommend authors make a heatmap for the frequencies of SARS-CoV-2 mutants, classify them into clusters based on the mutational signature by hierarchical clustering rather than just making stacking bar graphs. It is hard to understand that many colors.

Response: We greatly appreciate the reviewer input and suggestions. The suggested figure A1 is now incorporated within the revised manuscript.

  1. Though authors have extensively analyzed data by correlating single mutational to global CFR and deaths per million for figures 3, 4 and 5: Instead, it would be more informative and interesting to see correlations based on clusters or clades (like in figure6) obtained from hierarchical clustering which makes the entire manuscript simple and will be easy for the readers to understand

Response: We appreciate your clarification. Your suggestion is now incorporated in the updated version of the manuscript, Table 2 and Table 3.

  1. As D614G spike mutation in virus caused higher transmissibility and this mutation is detected in Europe in the early phase and has widely spread around the globe, especially to European and North American countries, it is obvious that it will be positively correlated with a number of cases per million and deaths per million.

Response: Thank you for your clarification. Totally agree with your clarification and this data was discussed in the discussion part.

  1. I recommend authors move to table 2 to supplementary data once the heatmap is made.

Response: We appreciate your suggestion. The table in the original manuscript was changed to Table 4, which was added to supplementary data in a revised version

Reviewer 2 Report

While I appreciate the authors' intent, I do not find the current study to be meritorious.

  1. The impact of the study, if successful, is questionable, given the data we now have on vaccines.
  2. The correlations are in large part driven by a few outlier countries in most cases
  3. There are very few data points, so the correlations are not compelling. They seem most easily explainable by chance associations.
  4. There are many confounding factors that could also explain the associations

In sum, I find the correlations to be questionable, and I also am not convinced (at this stage in the pandemic) that even true associations would have a major impact on our understanding of the virus.

Author Response

Response to Reviewers

We would like to thank the reviewers for their careful and thorough review of this manuscript and for the thoughtful comments and constructive suggestions, which help to improve the quality of this manuscript.

Please find attached a point-by-point response to the reviewer’s concerns. We hope that you find our responses satisfactory and that the manuscript is now acceptable for publication.

Response to Reviewer 2 Comments:

We appreciate your comments; we would like to clarify our point of view:

We believe that acquiring the gene sequences of our huge sample size may explain part of the differences of the SARS-CoV-2 infection rates, or the death rates, among various countries across continents which provides great medical value for future health care.

On the other hand, according to previous studies published in high-impact scientific journals. For example, Toyoshima et al. published a similar article in the Journal of Human Genetics using 12,343 SARS-CoV-2 genome sequences. In our study, we followed the same method and statistical analysis to analyze a huge sample size consisting of 553,518 SARS-CoV-2 genome sequences and correlated the results with CFR, Death/1M, and cases/1M of population.

The following are articles with similar themes/content published in high impact scientific journals

References

  1. Toyoshima Y, Nemoto K, Matsumoto S, Nakamura Y, Kiyotani K. SARS-CoV-2 genomic variations associated with mortality rate of COVID-19. Journal of human genetics. 2020 Dec;65(12):1075-82.
  2. Korber B, Fischer WM, Gnanakaran S, Yoon H, Theiler J, Abfalterer W, Hengartner N, Giorgi EE, Bhattacharya T, Foley B, Hastie KM. Tracking changes in SARS-CoV-2 Spike: evidence that D614G increases infectivity of the COVID-19 virus. Cell. 2020 Aug 20;182(4):812-27.
  3. Hamed SM, Elkhatib WF, Khairallah AS, Noreddin AM. Global dynamics of SARS-CoV-2 clades and their relation to COVID-19 epidemiology. Scientific reports. 2021 Apr 19;11(1):1-8.

Round 2

Reviewer 1 Report

Please check table numbers and relabel them properly 

This manuscript is a resubmission of an earlier submission. The following is a list of the peer review reports and author responses from that submission.

Round 1

Reviewer 1 Report

Overall, this is an interesting study that seeks to analyze and describe the effect of Covid19 virus mutations publicly available across the 6 continents. 

The largest weakness that must be addressed prior to publication is a clarification of the methods relating to statistical analyses between variant genomes and CFR. It appears as though the CFR is calculated across the various countries/continents and does not take into account the actual genetic strain of the virus. This is a major methodological error and should either be changed or made clear within the manuscript that the CFR does not link to specific sequences and is instead a population average.

Other analyses seem strong using publicly available data.

Author Response

Response to Reviewers

 We would like to thank the reviewers for careful and thorough reading of this manuscript and for the thoughtful comments and constructive suggestions, which help to improve the quality of this manuscript.

Please find attached a point-by-point response to reviewer’s concerns. We hope that you find our responses satisfactory and that the manuscript is now acceptable for publication.

Reviewer #1 (Comments to the Author): 

The largest weakness that must be addressed prior to publication is a clarification of the methods relating to statistical analyses between variant genomes and CFR. It appears as though the CFR is calculated across the various countries/continents and does not take into account the actual genetic strain of the virus. This is a major methodological error and should either be changed or made clear within the manuscript that the CFR does not link to specific sequences and is instead a population average.

Other analyses seem strong using publicly available data

Response: We appreciate your comments. Per your suggestion, we have clarified in the manuscript that the CFR was correlated to each of the respective mutations and not to specific virus strains. This has been incorporated in the updated version of the manuscript.

“Only high-coverage SARS-CoV-2 sequences were downloaded from GISAID EpiCoVTM database [12]. In particular, each of the SARS-CoV-2 sequences were compared with the reference sequence SARS-CoV-2 Wuhan-Hu-1 (accession number MN908947.3) by using CoVsurver enabled by GISAID (https://www.gisaid.org/epiflu-applications/covsurvermutations-app/) [12,13]. After the alignment of the full-length viral nucleotide sequence, we identified SARS-CoV-2 mutations in each of the isolated virus sequences. Single mutations based on a population average were used for the estimation of the case fatality ratio (CFR).”

Reviewer 2 Report

While I appreciate the question addressed by this manuscript, I have significant concerns about the interpretation of the results: 1. All of the significant correlations are driven by the Asian population. And in most cases the Asian population is a very large outlier in terms of variant frequency. A simple correlation statistic can be very biased by outliers. Are the correlations still significant if (a) a spearman correlation is run and (b) if the Asian population is removed? 2. Confounding factors could be driving these associations. What if Asian populations have lower CFR for a reason unrelated to genetics (e.g. better social distancing, better social responses) and have a higher mutation frequency for a different reason (e.g. the virus was established there earlier)? Then an association would be observed, but it would not be justified to conclude that the mutation was causal for the lower CFR. Again, confidence would be increased if an association was observed even among the non-Asian populations. But, latent confounders could still be present. The large fraction of associated variants (13/57, much higher than the 3 expected by chance), and the fact that all are driven by the Asian population, suggests that confounders are playing a large role here. 3. Even if the associations are real, another issue is data quality. What if Asian populations are undercounting cases? Then an association would not be biologically related at all. In summary therefore, the association methodology seems not robust to the current data, unmeasured confounders seem quite possibly driving the signals, and data quality issues could further confuse the results and their interpretation. I do find the paper very clear and well put together, so if these methodological concerns could be addressed I would view the paper favorably. Unfortunately I don't think data is available to do so. What variables did you control for? Regression of variant on CFR won't work because many latent factors for CFR that correlate with ancestry. Would need to analyze with ancestry at minimum but even that won't work Outliers could drive the signal 13/59 mutations with p<0.05 suggests associations are spurious Looks like being high in Asia is enough. What happens if we remove Asia? The Asia outlier is a huge problem -- all were from Asia What about reporting, measurement differences? Different from confounders; this could also mis-estimate

Author Response

Response to Reviewers

 We would like to thank the reviewers for careful and thorough reading of this manuscript and for the thoughtful comments and constructive suggestions, which help to improve the quality of this manuscript.

Please find attached a point-by-point response to reviewer’s concerns. We hope that you find our responses satisfactory and that the manuscript is now acceptable for publication.

Reviewer #2 (Comments to the Author): 

While I appreciate the question addressed by this manuscript, I have significant concerns about the interpretation of the results: 1. All of the significant correlations are driven by the Asian population. And in most cases the Asian population is a very large outlier in terms of variant frequency. A simple correlation statistic can be very biased by outliers. Are the correlations still significant if (a) a spearman correlation is run and (b) if the Asian population is removed? 2. Confounding factors could be driving these associations. What if Asian populations have lower CFR for a reason unrelated to genetics (e.g. better social distancing, better social responses) and have a higher mutation frequency for a different reason (e.g. the virus was established there earlier)? Then an association would be observed, but it would not be justified to conclude that the mutation was causal for the lower CFR. Again, confidence would be increased if an association was observed even among the non-Asian populations. But, latent confounders could still be present. The large fraction of associated variants (13/57, much higher than the 3 expected by chance), and the fact that all are driven by the Asian population, suggests that confounders are playing a large role here. 3. Even if the associations are real, another issue is data quality. What if Asian populations are undercounting cases? Then an association would not be biologically related at all. In summary therefore, the association methodology seems not robust to the current data, unmeasured confounders seem quite possibly driving the signals, and data quality issues could further confuse the results and their interpretation. I do find the paper very clear and well put together, so if these methodological concerns could be addressed I would view the paper favorably. Unfortunately I don't think data is available to do so. What variables did you control for? Regression of variant on CFR won't work because many latent factors for CFR that correlate with ancestry. Would need to analyze with ancestry at minimum but even that won't work Outliers could drive the signal 13/59 mutations with p<0.05 suggests associations are spurious Looks like being high in Asia is enough. What happens if we remove Asia? The Asia outlier is a huge problem -- all were from Asia What about reporting, measurement differences? Different from confounders; this could also mis-estimat

Response: Thank you for the insightful comments and observations. Per your suggestions, we performed Spearman’s correlation for the mutations. This led to obtaining no significant correlation for 11 mutations that were originally negatively correlated with the CFR using Pearson Correlation and obtaining a new positive correlation between N_G204R and CFR. This meant that the Data from Asia were outliers. Therefore, data from Asia were removed. This led to a decrease in the number of total mutations from 59 to 51 because 8 of the reported mutations were unique to Asia. Hence, we performed Spearman’s correlation analyses on the dataset after excluding data from Asia. This new analysis revealed two positive correlations between each of N_G204R and N_R203K with CFR and two negative correlations with each of NSP2_R380H and NS3_Q57H_A with CFR. Other parts of the manuscript were revised to address changes in the results. Also, the methods section was revised to indicate that “Spearman’s Correlation” was utilized instead of “Pearson Correlation”.

Round 2

Reviewer 2 Report

I appreciate that the reviewers have addressed my earlier comments. I think unfortunately that the new data still does not appear to support any strong associations between variant frequencies and case fatality ratio. The p-values of the four variants are not significant after correcting for 50 tests. In addition, it's pretty clear to me from Figures 3 and 4 that the two positively correlated variants follow exactly the same pattern, as do the two negatively correlated variants, which suggests they are most likely products of the same ascertainment issues as opposed to having any causal effects on CFR.